



# Study on the drought risk of maize in the farming-pastoral ecotone in Northern China based on physical vulnerability assessment

Zhiqiang Wang[1,2], Jingyi Jiang[2,3], Qing Ma[4,5]

[1]National Disaster Reduction Center/Satellite Application Center for Disaster Reduction of the Ministry of Civil Affairs, Beijing 100124, China

[2]Key Laboratory of Disaster Reduction and Emergency Response, Ministry of Civil Affairs, Beijing 100124, China

[3]INRA-EMMAH UMR 1114, 84914 Avignon, France

4College of Geography, Beijing Normal University, Beijing 100875, China

[5]Chongqing No.18 High School, Chongqing 400020, China

*Correspondence to*: Jingyi Jiang (jiangjingyi1989@163.com)

**Abstract.** Climate change is affecting every aspect of human activities, especially the agriculture. In China, extreme drought events caused by climate change have posed great threaten to food safety. In this work we aimed to study the drought risk of maize in the farming-pastoral ecotone in Northern China based on physical vulnerability assessment. The physical vulnerability curve was constructed from the relationship between drought hazard intensity index and yield loss rate. The risk assessment of agricultural drought was conducted from the drought hazard intensity index and physical vulnerability curve. Results of the drought hazard intensity index showed that the risk of agricultural drought displayed a negative correlation with the precipitation and kept rising from 1966 to 2011. Risk assessments of yield loss ratio shows that physical vulnerability curve has magnify and reduce function to drought hazard. So improving the capacity of maize to resist drought can help them adapt to drought hazard. In conclusion, the farming-pastoral ecotone in Northern China had great sensitivity to climate change and high probability for severe drought hazard. Risk assessment of physical vulnerability can help better understanding the physical vulnerability to agricultural drought and can also promote measurements to adapt to the climate change.

## 1 Introduction

Climate change and its influence on human activity have gained more and more attention from different fields and communities. In the past 30 years, the global surface temperature kept the linear growth trend





and the frequency and intensity of extreme climate events increased (IPCC, 2014). Even though many

measures have been taken to response to climate change, the tendency will keep in the next period of

time. In arid areas, water shortage resulted from climate change during the crop growing stage will lead

to loss of yield, increase the frequency of agricultural drought and threaten the food security (FAO,

2013;Wheeler and von Braun, 2013). In Africa, climate change will amplify existing stress on water

availability and agricultural systems particularly in semi-arid environments. In Asia, agricultural

productivity will be declined because of the climate change in many sub-regions for crops like rice

(IPCC, 2014). For China, drought is one of the most obvious performances of climate change (Piao et al.,

2010). In the past 60 years, China has suffered a number of severe drought hazards which caused great

loss of agricultural production (Zou et al., 2005). So better understanding and evaluation of agricultural

drought hazard can help people raise the capacity of responding to agricultural drought hazard and put

forward countermeasures to reduce risk of agricultural drought hazard in high-risk areas.

As the core of disaster risk assessment, study about vulnerability has been widely applied in many

different fields like ecology, public health and global climate change (Füssel, 2007). Initially,

vulnerability was defined as the human response to hazard events (FAO, 2001;Blaikie and Cannon,

1994). Gradually, vulnerability is added with some new meanings including the different systems of

human society responding to hazard, the interaction process of multi-factors like nature, society,

economy and environment (UNDP, 2004) , the sensitivity or susceptibility to hazards and the capacity to

cope and adapt to hazards (IPCC, 2014). Many methods have been proposed to assess vulnerability.

Statistical method builds the relationship between impact factors and vulnerability to reflect the

characteristic of vulnerability (Simelton et al., 2009). But this method cannot consider different kinds of

factors synthetically. Another kinds of methods like fuzzy modeling (Alcamo et al., 2005;Azadi et al.,

2009) and multi-indicator method (Antwi-Agyei et al., 2012;Kim et al., 2013) can provide a compressive

assessment for different impact factors. But fuzzy modeling method is restricted by background

knowledge and information. So it is hard to set weight for different indicators. For multi-indicator

method, some information will be dropped during the process of integrating different indicators. Cluster

analysis method can consider different impact factors separately and determine the most vulnerable

places based on different combinations of the components of vulnerability (Sharma and Patwardhan,

2008). The advantage of this method is its ability to maintain information from different factors

completely. Compared with fuzzy modeling and multi-indicator method, it can provide richer


information for decision makers. But this method is always restricted to qualitative description of

vulnerability and cannot quantitatively describe the vulnerability of hazard-affected body.

However, all those studies discussed above did not consider physical factors and social factors

separately. This will cause great restrict to realize the mechanism of disaster from its formation process.

Different from the system vulnerability caused by social factors, physical vulnerability is an internal

characteristic of hazard-affected body. It is the capacity of hazard-affected body to response, resist and

recover from strike caused by nature or human beings (Wang et al., 2013). Many studies have

quantitatively analyzed the physical vulnerability of different disasters: Uzielli et al. (Uzielli et al., 2008)

utilized the relationship between landslide intensity and the susceptibility of vulnerable elements to

quantitatively estimate physical vulnerability of the built environment and population to landslides.

Douglas (Douglas, 2007) used fragility curves to model physical vulnerability for evaluation of

earthquake and landslide risk. For physical vulnerability of agricultural drought hazard, most studies

chosen geographical statistical methods or different drought index with meteorology, hydrology or

remote sensing data to calculate the distribution of drought hazard risk (Murthy et al., 2015;Kellner and

Niyogi, 2014;Karavitis et al., 2014;Jain et al., 2015). But all these studies did not consider the

relationship between meteorological factor and crop water stress during the process of crop growth, so it

is hard to determine the physical vulnerability of crop to drought hazard. Based on the crop growth

model, Wang et al. (Wang et al., 2013) proposed a Hazard–Loss curve to construct the relationship

between drought hazard intensity and crop yield loss and utilized the curve to simulate the physical

vulnerability of hazard-affected crop with environmental policy-integrated climate (EPIC) model. As the

drought hazard intensity is calculated from the accumulation of crop daily water stress, so the physical

vulnerability curve can better reflect the biophysics regulation during crop growth and avoid errors

caused by the integration of multi-parameters. Physical vulnerability curve makes the drought hazard

vulnerability become a parameter which can be described quantitatively in a dynamic process and

provides the probability to assess physical vulnerability of agricultural drought hazard from the aspect of

disaster mechanism.

To construct physical vulnerability curve, the key point is to calculate the drought hazard intensity

index and the corresponded crop yield loss ratio. In previous study (Wang et al., 2015),a new method

was proposed to determine drought hazard intensity index based on the daily water stress from EPIC

model and yield loss contribution rates for different growth stages. In this study, the risk assessment of





agricultural drought was conducted from the physical vulnerability curve. Firstly, under the circumstances of no irrigation, the drought hazard intensity index was calculated from the daily water stress and yield loss contribution rates for different growth stages. Based on the distribution of drought hazard intensity index, the risk of drought hazard intensity index in different regions was analyzed. Then, the yield loss ratio was got from the difference of yield with two different scenarios (sufficient irrigation

and no irrigation). With the spatial distribution of drought hazard intensity index, sites with different drought hazard intensity index and yield loss ratio were selected. A Logistc model was used to simulate the physical vulnerability curve of crop from the relationship between hazard and loss. According to the physical vulnerability curve, both the physical vulnerability assessment and risk assessment of yield loss ratio were analyzed. As the farming-pastoral ecotone in Northern China is located at the region which is

sensitive to climate change and belongs to rain-fed agriculture region which is fragile in ecology, agriculture drought hazard risk assessment of this region can reflect the time-spatial variation of drought hazard resulted from climate change. So the farming-pastoral ecotone in Northern China was chosen as the study area in this study. In addition, as the spring maize, which is a kind of drought-resistant crops, is widely planted in this region, so spring maize was selected as a typical crop for the risk assessment of

physical vulnerability in the farming-pastoral ecotone in Northern China. The assessment results showed the farming-pastoral ecotone in Northern China is a region with high risk of agricultural drought. To better adapt to drought, more attentions should be paid in this region and more methods like changing the growth environment of crop to reduce the strength of drought hazard intensity index, developing improved varieties of crops to reduce physical vulnerability of agricultural drought and reducing crop's

exposure to drought during the planting process should be adopted to responses to climate change.

**2 Data and methodology**

**2.1 Study area and data**

**2.1.1 Study area**

Ecotone is defined as a multi-dimensional environmentally interaction zone between ecological systems

(Hufkens et al., 2009). Because of its sensitivity to climatic variation, environmental change and human activity, ecotone tends to shift in space and time over several spatial scales (Kark, 2013). For example, the African Sahel is regarded as a typical ecotone, which is influenced by fluctuations in climate and



human activities (Herrmann et al., 2005;Rian et al., 2009). In East Asia, the farming-pastoral ecotone in

Northern China stretching across the monsoon fringe area from southwest to northeast are dominated by

adjacent ecological systems of steppes and crops (Lu and Jia, 2013). As the typical ecotone with the

largest area and longest span in the world, the farming-pastoral ecotone in Northern China is highly

sensitive to climate change in East Asia. Many researchers utilized historical meteorological data

including temperature, precipitation and remote sensing data to compositely analyze the impacts of

climate change and socioeconomic factors on boundary shifts and land use change of the

farming-pastoral ecotone in Northern China (Liu et al., 2011;Ye and Fang, 2013;Geng et al., 2014;Shi et

al., 2014). Results show that the extent of the farming-pastoral ecotone in Northern China greatly

fluctuated in accordance with the variability of precipitation (Lu and Jia, 2013;Geng et al., 2014).

Meanwhile, the dry climate conditions and long-term excessive human activity have made desertification

in this region a serious environmental problem that affects the economy and society development (Xu et

al., 2014). Based on the simulation from land use scenario dynamic model, it is predicted that the

farming-pastoral ecotone in northern China will become increasingly vulnerable and hotspots for

land-use change with the intensified drying trends (Geng et al., 2014).

There exist many different definitions about the farming-pastoral ecotone in northern China. In

general, it is located at the north part of China with the rainfall isoline changing from 300mm to 400mm,

annual precipitation change ranging from 15% to 30% and dryness changing from 1.0 to 2.0 (Zhao et al.,

2002). The precipitation period is between June and August with large interannual variation. According

to the distribution of landform and vegetation zonality, the farming-pastoral ecotone in northern China

can be classified into three parts: the east part, the middle part and the west part. The east part is the

transition area of Northeast China Plain and Inner Mongolian Plateau with vegetation types changing

from warm temperate deciduous forests to temperate forest steppe. The annual average temperature is

from 3 to 7℃.The middle part transits from North China Plain to Loess Plateau with the vegetation

changing from warm temperate deciduous forests to temperate grassland. Because of the relatively high

altitude, the annual average temperature here is from 0 to 1℃. The west part is the transition area from

Loess Plateau to Qinghai-Tibet Plateau with the vegetation changing from warm temperate deciduous

forests to temperate desert steppe. The annual average temperature is from 6 to 9℃. Fig. 1 shows the

location and the land-use of the farming-pastoral ecotone in Northern China. The east part, middle part

and west part of the farming-pastoral ecotone in Northern China are located from east to west



respectively. The blue triangles represent 54 meteorological stations covering the whole region and the

yellow regions are cultivated land. Different shades of black lines show the change of rainfall isolines,

which increase from northwest to southeast and extension alone the direction of northeast to southwest.

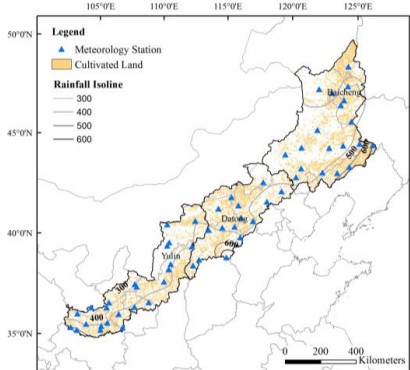

**Figure 1: Location and the land-use map of the farming-pastoral ecotone in Northern China**

**2.1.2 Data**

The data used in this study include meteorological data of the farming-pastoral ecotone in Northern

China from 1966 to 2011, soil data and the agricultural data (Table 1). The daily meteorological data

came from 54 meteorological stations of the study area. The soil texture data were retrieved from

''Chinese Soil Genus Records'' and transformed into the USDA soil-type system (Skaggs et al., 2001).

**Table 1: Meteorological data, soil data and relative agricultural data**

| Database | Content | Source |
|---|---|---|
| Meteorological data | Daily data from 1960 to 2011: precipitation, temperature, radiation, wind speed and relative humidity et al. | China meteorological data sharing service system of China Meteorological administration |
| Spatial distribution of soil | 1:1,000,000 soil map of the study area | Institute of Soil Science, Chinese Academy Sciences |
| Soil properties | Soil layers, texture data and organic carbon and so on | Chinese Soil Genus Records |
| Land use map | Land use map of the study area in 2000. The main land use type including paddy field, dry land, forest land, grass land and so on. | Institute of Remote Sensing and Digital Earth Chinese Academy of Sciences |
| Statistical agricultural data | Fertilization, sowing area, yields from 1966 to 2011. | China Statistical Yearbook |

**2.2 Methodology**

The process of assessment is shown in Fig. 2. As the disaster risk (R) are the function of hazard factor



(H), physical vulnerability (V), exposure (E) and the disaster reduction capacity of hazard-affected body

to hazard (CH), physical vulnerability (CV) and exposure (CE) (Eq. 1), so under the assumption that all

maize was exposed to drought hazard, both hazard intensity index and the physical vulnerability curve

was simulated through EPIC model to assess the risk of spring maize in the farming-pastoral ecotone in

Northern China.

$$R = \frac{H \times V \times E}{C_H \times C_V \times C_E}$$

(1)

The station EPIC model was used to calibrate the genetic parameter of local maize. Then a

water-deficit experiment on different growth stage was conducted to calculate the yield loss contribution

rate. So the maize drought hazard intensity index was defined based on daily water stress and the yield

loss contribution rate. Added with the spatial data in time series, the spatial EPIC model was used to

calculate the spatial maize drought hazard intensity index. Meanwhile, the yield loss ratio was got from

the difference of yield under two different simulated scenarios with EPIC model (One was sufficient

irrigation and the other one was no irrigation). After this, the physical vulnerability curve was

constructed from sites under different drought hazard intensity using Logistic regression model to

describe the relationship between drought hazard and yield loss. According to the function of disaster

risk, the risk assessment of drought hazard was conducted from three aspects: the first was the risk

assessment of drought hazard intensity index; the second was physical vulnerability assessment of spring

maize based on physical vulnerability curve; the third one was the risk assessment of yield loss ratio

calculated from physical vulnerability curve.





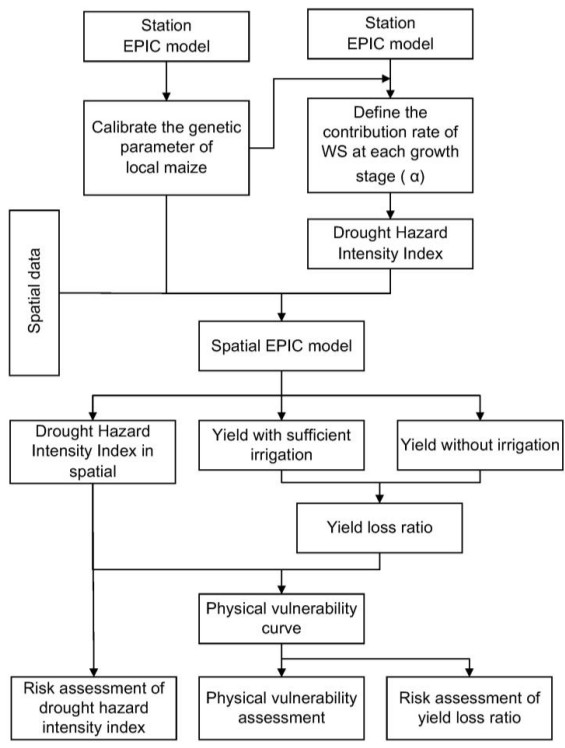


**Figure 2: Flow chat of drought risk based on physical vulnerability assessment**

**2.2.1 Drought Hazard Intensity Index**

EPIC model is a field-scale crop model, which is capable of simulating daily crop growth, calculating

crop yield under various climate and environment conditions and performing long-term simulations for

hundreds of years (Gassman et al., 2005). For different kinds of crop, genetic parameters used in EPIC

model vary with different varieties and geographical conditions. So before the simulation with EPIC

model, it is necessary to localize the genetic parameter of crops based on the field measurements. In

addition, parameters like soil parameters, filed management parameters and daily meteorological data are

also input parameters to run EPIC model.

190       According to the three parts of the farming-pastoral ecotone in Northern China defined in the last

section, we chose three represent stations (east part: Baicheng, middle part: Datong, west part: Yulin) to

calculate genetic parameters of spring maize of each part. In each station, the annual yields of spring

maize from 2000 to 2005 from agriculture statistic yearbooks were selected as the recorded yields to



adjust the genetic parameters. In addition, the daily meteorological data from 2000 to 2005, the soil data

and the field management data were all put into the station EPIC model. The genetic parameters of the

spring maize were finally determined after a number of adjustments based on the comparison between

the model output crop yields and the recorded data. The simulation results of spring maize for each

station are demonstrated in Fig. 3. To validate the accuracy of the determined genetic parameters, the

annual yields from agriculture statistic yearbooks in another 6 stations within the study area from 2000 to

2005 were selected. Fig. 4 shows the validation result between the simulated results and the recorded

yields. The correlation coefficient $R^2$ is 0.86. Seen from the validation result, the determined genetic

parameters of spring maize were appropriable for this study. For the difference between simulated results

and recorded data, errors mainly came from the input data of station EPIC model including daily

meteorological data and yield management data.


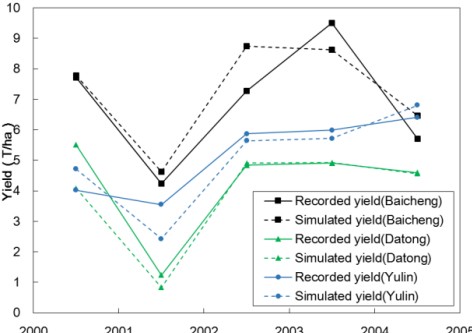

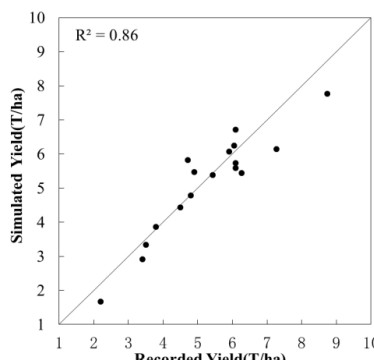

Figure 3: Calibration of the genetic parameters using the recorded yields of spring maize at Baicheng, Datong, Yulin station.

Figure 4: Validation of the genetic parameters using the recorded yields of spring maize at six stations in the study area from 2000 to 2005.

After running the EPIC model, parameters which describe the growth state of crops are output.

Among them, water stress, which can reflect the relationship between water supply and demand during

the crop growth process, is an important parameter in risk assessment of agricultural drought. So during

the calculation of drought hazard intensity index, water stress is selected as the main factor to describe

the intensity of drought. In addition, because water stress will have different influence on crop yield in

different crop growing stage, so the yield loss contribution rate of water stress ( $\alpha$ ) (Wang et al., 2015),

which is calculated from the proportion of yield reduction rate in each stage to the summary of the yield

reduction rate in different stages (Eq. 2), is also used to reflect water critical period for crop yields. *m*





stages of crop growth are defined based on crop growth regulation.

$$\alpha_i = \frac{y_i}{\sum_{i=1}^{m} y_i}$$

(2)

where $y_i = Y_i/Ys$, $Ys$ is the yield under the situation of sufficient soil water supply and $Yi$ is the yield under

water shortage in growth stage $i$. m is the number of growth stages.

Based on the EPIC water stress and the calculated yield loss contribution rate, the equation of

drought hazard intensity index is as follows (Wang et al., 2015):

$$DHI_{yj} = \frac{\sum_{k=1}^{n} \alpha_i (1 - WS_k)}{\max DHI}$$

(3)

where $DHI_{yj}$ is the drought hazard intensity index of y year in unit $j$; $WS_k$ is to the water stress on day $k$;

$\alpha_i$ is the yield loss contribution rate of $WS_k$ and $k$ belongs to the $i$ growth stage; $n$ is the total days when

there is water stress; max $DHI$ denotes the maximum value of $\sum_{k=1}^{n} \alpha_i (1 - WS_k)$ in all the simulated

years and units. $DHI_{yj}$ varies from 0 (0 represents the minimum intensity of drought hazard) to 1(1

represents the maximum intensity of drought hazard).

### 2.2.2 The establishment of physical vulnerability curve

To determine the impact of water stress on crop yields, both the crop genetic parameters and daily

meteorological data after spatial interpolation are put into the spatial EPIC model. With the guarantee of

soil nutrient and ventilation, two scenarios (*Y1*: sufficient irrigation; *Y2*: no irrigation) are set to simulate

the growth of the crop. For each unit, the difference of crop yields between two scenarios for each year is

the year yield loss under water stress. So the proportion between the year yield loss and the maximum

yield with sufficient irrigation for multi-years in this unit is defined as the yield loss ratio under water

stress (Wang et al., 2013):

$$YL_{yj} = \frac{Y1_y - Y2_y}{\max Y1_j}$$

(4)

where $YL_{yj}$ is the loss rate of yield of $y$ year in station $j$; $Y1_y$ and $Y2_y$ are the unit yield of $y$ year under

scenario *Y1* and *Y2*; max $Y1_y$ represents maximum unit yield of station $j$.

Based on drought hazard intensity index and the corresponded yield loss rate, the physical

vulnerability curve is defined to simulate the relationship between them using the regression analysis



method. For each kind of crop, the physical vulnerability curve is an internal property for hazard-affected

body itself. The key point to establish the physical vulnerability curve is to set up a wide range of

sceneries from no drought to extreme drought hazard. The more scenarios of drought hazard intensities

are included, the more accurate physical vulnerability curve will be got. So in this study, according to the

drought hazard intensity index we calculated from different evaluation units in different years, we try to

select more sites under different sceneries of drought hazard intensities. And then the Logistic regression

analysis is used to simulate the physical vulnerability curve of the crop from selected points.

### 1.2.3 Risk assessment of agricultural drought

In this study, drought hazard intensity index and physical vulnerability curve are two cores of the drought

risk assessment. Based on the formulation process of agricultural drought hazard, the risk assessment of

drought hazard intensity index, the physical vulnerability assessment on spring maize and the risk

assessment of yield loss ratio calculated from physical vulnerability curve were conducted sequentially.

For drought hazard intensity index, as it is calculated from crop daily water stress and yield loss

contribution rate, it can reflect the degree of agricultural drought on specific crop. So the time series,

standard deviation and slope of drought hazard intensity index on each evaluation unit were calculated to

analyze the spatial-temporal distribution of agricultural drought. The probability distribution of drought

hazard intensity index was processed for the risk assessment. For physical vulnerability assessment, it

was conducted relying on physical vulnerability curve of spring maize. For the yield loss ratio calculated

from physical vulnerability curve, as it is determined by both drought hazard intensity index and the

corresponded physical vulnerability, so the calculated yield loss ratio is a good represent of drought risk.

The standard deviation, slope and probability of yield loss ratio were calculated to show the

spatial-temporal distribution and probability distribution of drought risk.

### 3 Results

### 3.1 Risk assessment of maize drought hazard intensity index

Based on the classification method of annual crop climate types (AQSIQ/SAC, 2008), year 2002 was

selected as the climate normal year to calculate the yield loss contribution rates of water stress at

Baicheng, Datong and Yulin with the corresponded genetic parameters of the spring maize. According to



the growth regulation of spring maize, six growth stages were determined in Table 2. With the crop yield under sufficient irrigation as the comparison, the water-deficit treatment in each growth stage was conducted respectively with station EPIC model. The resulted yield losses were recorded. According to Eq. 2, the yield loss contribution rates in different growth stages for each station were shown in Table 2.

For more details about the experiment of water-deficit, Wang et al. (Wang et al., 2015) is referenced.

**Table 2: Yield loss contribution rate α at Baicheng, Datong and Yulin.**

| | | Growth stage of spring maize | | | | | | |
|---|---|---|---|---|---|---|---|---|
| | | Seeding stage | Jointing stage | Early heading stage | Late heading stage | Early filling stage | Late filling stage | Mature stage |
| Yield loss contribution Rate $\alpha_i$ | Baicheng | 0.17 | 0.22 | 0.18 | 0.18 | 0.16 | 0.09 | 0 |
| | Datong | 0.16 | 0.25 | 0.19 | 0.16 | 0.15 | 0.08 | 0 |
| | Yulin | 0.17 | 0.18 | 0.22 | 0.17 | 0.16 | 0.10 | 0 |

We used the genetic parameters and yield loss contribution rates got from three sites to represent genetic parameters and yield loss contribution rates in the east, middle and west part of the farming-pastoral ecotone in Northern China respectively. With the evaluation unit of 5km×5km area, the

interpolated meteorological data from 1966 to 2011 and the genetic parameters for each part were put into the spatial EPIC model. Under the situation of no irrigation, the daily water stress of each evaluation unit in study area was output. Based on Eq. 3, the spatial distribution of drought hazard intensity index of the farming-pastoral ecotone in Northern China was calculated finally.

Fig. 5 shows the distribution of spring maize drought hazard intensity index in the farming-pastoral

ecotone in Northern China in every five year. For most years, the drought hazard intensity index of spring maize increased from northeast (0.1) to southwest (0.5) and decreased back to 0.1 at the margin of southwest. Compared with the rainfall isoline, there existed a negative correlation between the drought hazard intensity index and precipitation. For regions with the rainfall isoline lower than 300mm, drought hazard intensity index for most years were around 0.5 or 0.6. But with the rise of precipitation, drought

hazard intensity index declined gradually. For regions with rainfall isoline from 500mm to 600mm, most drought hazard intensity indexes centered on 0.1. In generally, the middle part and most region of the west part were the driest part of the whole study area with the average of drought hazard intensity index for multi-years larger than 0.5.

Seen from the time series of drought hazard intensity index in the farming-pastoral ecotone in




Northern China, there existed two extreme drought hazards in 1980 and 2000 with the average of drought

hazard intensity index larger than 0.8. For other years, areas with the drought hazard intensity index

larger than 0.5 mainly centered in the west part. But since 2005, regions with the drought hazard intensity

index larger than 0.5 spread toward northeast. In 2010, except little areas, the drought hazard intensity

index for the whole study ranged from 0.4 to 0.6.

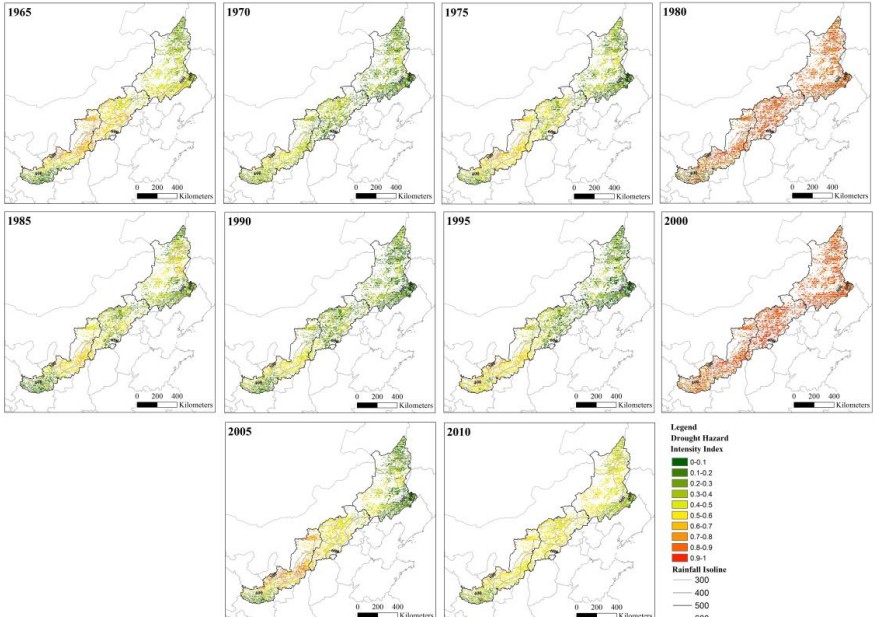


**Figure 5: The spatial distribution of spring maize drought hazard intensity index in time series.**

        To reveal the fluctuation of drought hazard intensity index of spring maize in time series, the

standard deviation from 1966 to 2011 of each evaluation unit in the study area was calculated. Fig. 6 is

the distribution of standard deviation of spring maize drought hazard intensity index from 1966 to 2011.

For most parts, the standard deviation was from 0.1 to 0.3 and had the tendency of decreasing from

northeast to southwest. The east part had the highest standard deviation (0.3), which showed greatest

interannual fluctuation of spring maize drought hazard intensity index. But the standard deviation in west

part ranging from 0.1 to 0.2 was relatively low.

        In order to describe the variation tendency of drought hazard intensity index from 1966 to 2011 in

the farming-pastoral ecotone in Northern China, the slope of linear regression of drought hazard intensity

index for 46 years were calculated. Fig. 7 is the distribution of slope of spring maize drought hazard

intensity index from 1966 to 2011. Warm-toned colors like red and yellow represent slope larger than 0




and the increasing tendency of drought hazard intensity index, while cool-toned colors like green represent slope smaller than 0 and the decreasing tendency of drought hazard intensity index. For the

whole farming-pastoral ecotone in Northern China, the change of slope was small ranging from -0.006 to 0.006. But for the most parts, the slope was larger than 0, which meant the intensity of drought hazard strengthened. The middle part had the most obvious increasing tendency (from 0.002 to 0.006).The next was east part within the rainfall isoline from 300mm to 400mm and little region at the southwest edge of the study area with the slop changing from 0.002 to 0.004.

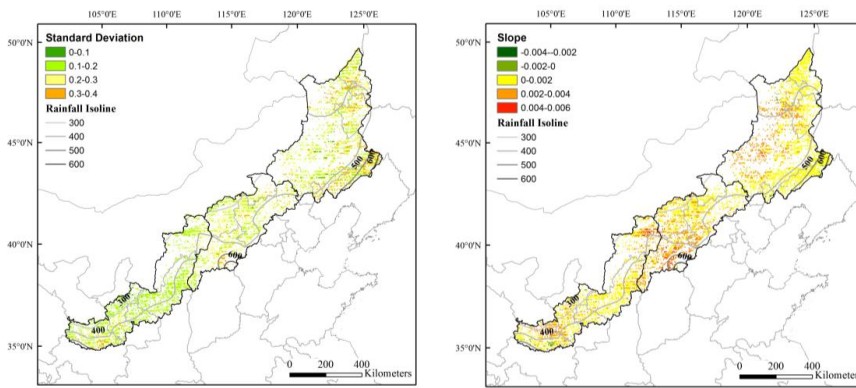

**Figure 6: Standard deviation of spring maize drought hazard intensity index from 1966 to 2011.**

**Figure 7: Slope of linear regression of spring maize drought hazard intensity index from 1966 to 2011.**

To access the risk of spring maize drought hazard intensity index in the farming-pastoral ecotone in Northern China, we calculated the exceeded probability of spring maize drought hazard intensity index for each evaluation unit. Through fixing the drought hazard intensity index, probability distribution of spring maize drought hazard intensity index with 4 hazard levels was drawn separately: spring maize drought hazard intensity index≥0.2 (Fig. 8(a)), spring maize drought hazard intensity index≥0.3 (Fig.

8(b)), spring maize drought hazard intensity index≥0.4 (Fig. 8(c)) and spring maize drought hazard intensity index≥0.5 (Fig. 8(d)).

        Seen from Fig. 8, the high value area of the probability was located at most region of west part and middle part, while the low value area was located at the east part within the rainfall isoline from 500mm to 600mm and the southwest edge of the west part. For most region of west part, the upper limit of

probability was 1 under four hazard levels. So there existed very high probability of drought hazard and would encounter big disaster losses almost every year. For the middle part, the upper limit of probability under 4 hazard levels from 0.2 to 0.5 was 1, 1, 0.8 and 0.5 respectively. So in this region, drought hazard



with the intensity index of 0.2 and 0.3 would occur nearly every year. The risk level of drought hazard

with the intensity index of 0.5 was every two years. For these two low value regions, the upper limit of

probability under 4 hazard levels from 0.2 to 0.5 was 1, 0.9, 0.4 and 0.2 respectively. These regions

would meet the drought hazard with the intensity index of 0.2 every year and the risk level of drought

hazard with the intensity index of 0.5 was at least every five years.

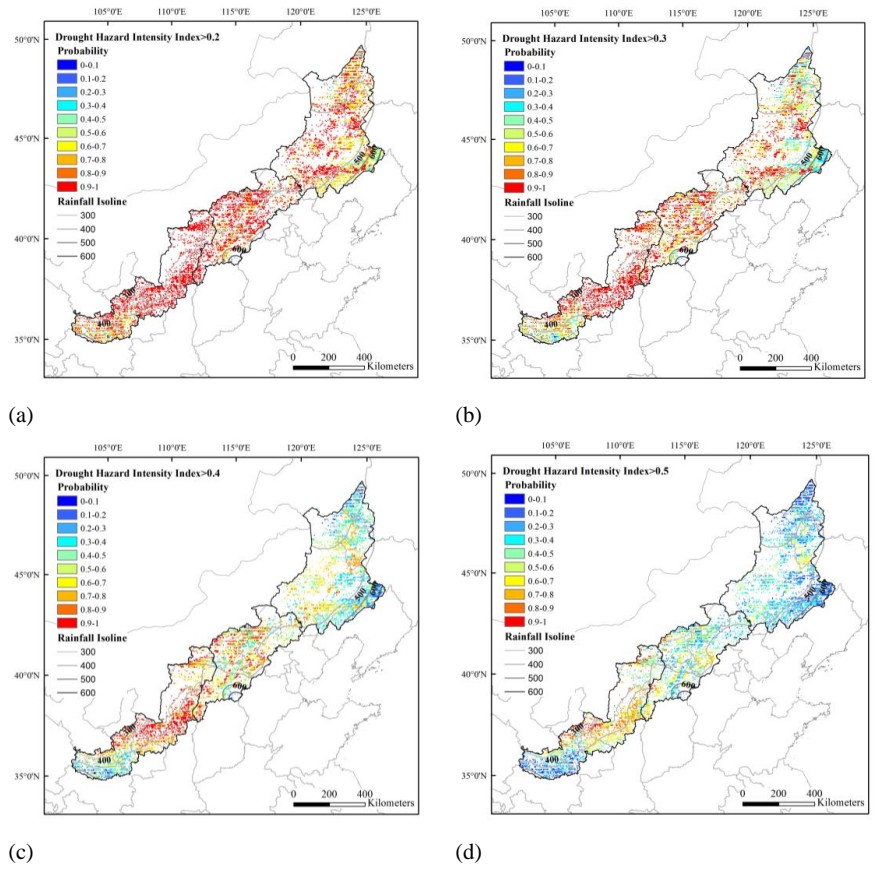

**Figure 8: Probability distribution of spring maize drought hazard intensity index for different hazard levels.**
**(a) Drought hazard intensity index≥0.2; (b) Drought hazard intensity index≥0.3; (c) Drought hazard intensity**
**index≥0.4; (d) Drought hazard intensity index≥0.5.**

**3.2 Physical vulnerability curve of spring maize**

Based on the genetic parameters of spring maize for three parts of farming-pastoral ecotone in Northern

China, the spatial EPIC model was conducted under two scenarios ($Y1$: sufficient irrigation; $Y2$: no

irrigation). For each evaluation unit, the loss rate of yield was got according to Eq. 4. Combined with the

spatial distribution of drought hazard intensity index, 50 sites under different drought hazard intensity



were selected for each part to extract drought hazard intensity index and the corresponded loss rate of yield for 46 years. For each part of the farming-pastoral ecotone in Northern China, a scatter diagram which describes the drought hazard intensity curve-yield loss rate is shown in Fig. 9. Each point represents the annual loss rate of yield and drought hazard intensity for a given site. The solid line is the

physical vulnerability curve simulated using logistic curve fitting methodology. For each part, the physical vulnerability of spring maize could be evaluated using the physical vulnerability curve as follows:

$$Y_e = \frac{0.57}{(1+8.81e^{-6.33H_e})} \tag{5}$$

$$Y_m = \frac{0.58}{(1+11.49e^{-8.9H_m})} \tag{6}$$

$$Y_w = \frac{0.54}{(1+10.821e^{-7.68H_w})} \tag{7}$$

where $Y_e$, $Y_m$ and $Y_w$ represent the yield loss ratios of spring maize in east, middle and west parts and $H_e$, $H_m$ and $H_w$ are the drought hazard intensity indexes in east, middle and west parts. R² is 0.70 for east part, 0.71 for middle part and 0.65 for west part.

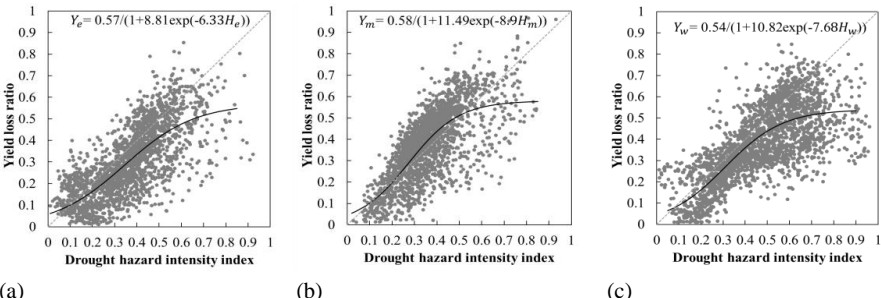

(a)                          (b)                          (c)

**Figure 9: Physical vulnerability curve to drought hazard of spring maize at (a) east part (b) middle part and (c) west part of the farming-pastoral ecotone in Northern China.**

Restricted by the meteorological data, it was hard to include every different meteorological scenery in theoretical like extreme drought (H = 1) or no drought (H = 0) to simulate a real physical vulnerability curve to drought hazard of spring maize. In addition, errors from meteorological data and the model itself

might also have impacts on simulation results. So considering the accuracy of the input data and some uncertainties during the calculation process, the simulated drought physical vulnerability curve of spring maize for each part was satisfied in this study.



Comparing three different kinds of spring maize planted in the study area, the physical vulnerability curve of spring maize for each part is slightly different from each other. But as these three regions are adjacent and are all located at the farming-pastoral ecotone in Northern China which is relatively drought, the difference is not obvious. For each curve, at the beginning stage, the yield loss ratio is mild with low drought hazard intensity index (0 to 0.2). Then the increase of yield loss ratio is swift with middle drought hazard intensity index (0.2 to 0.6). For the last stage, the yield loss ratio reaches the highest point and becomes stable with high drought hazard intensity index (0.6 to 1). For each part of the study area, all of these curves are below or near 1:1 line, which show reduction-effect of drought hazard intensity and the reduction of drought hazard vulnerability on spring maize.

### 3.3 Risk assessment of yield loss ratio based on physical vulnerability curve

We assumed that the spring maize in the study area was completely exposed to drought hazard, so the drought risk of spring maize is mainly determined by the drought hazard intensity index and its vulnerability under this drought hazard intensity. So in this section, we put the calculated drought hazard intensity index of each evaluation unit into the physical vulnerability curve to get the simulated yield loss ratio. The risk assessment was conducted on the basis of the distribution of yield loss ratio for each unit.

To describe the change of yield loss ratio in time series, both the standard deviation and the slope of yield loss ratio from 1966 to 2011 were calculated on each evaluation unit. Fig. 10 is the distribution of standard deviation of yield loss ratio in farming-pastoral ecotone in Northern China. Similar with the standard deviation of drought hazard intensity index, the standard deviation of yield loss ratio also showed the tendency of decline from northeast to southwest. But because of the reduction-effect of the physical vulnerability curve, the standard deviation of yield loss ratio was slightly lower than that of drought hazard intensity index. Here the standard deviation of yield loss ratio for east part was from 0.1 to 0.3. For middle and west part, it was ranging from 0 to 0.2.So the interannual instability of yield loss was reduced. Fig. 11 is the slope of linear regression of yield loss ratio. Slope larger than 0 is showed with warm-toned colors and represents the increasing tendency of yield loss ratio, while slope smaller than 0 is showed with cool-toned colors and represents the decreasing tendency of yield loss ratio. The same increasing tendency like slop of drought hazard intensity index was demenstrated. But impacted by the reduction-effect of the physical vulnerability curve, the rising trend was slightly lower compared with the slope of drought hazard intensity index. Here the highest values of slope of yield loss ratio (exceeding




0.004) centered on middle part, while the other parts changed from 0 to 0.002.

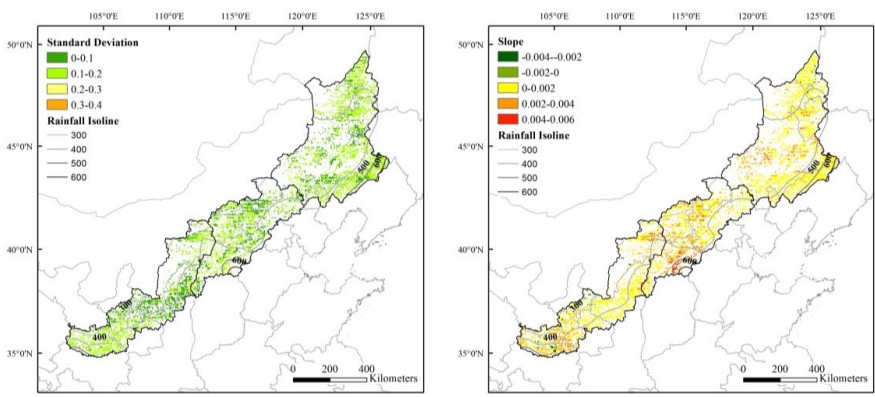

Figure 10: Standard deviation of spring maize yield loss ratio from 1966 to 2011.

Figure 11: Slope of linear regression of spring maize yield loss ratio from 1966 to 2011.

To assess the risk of yield loss ratio based on physical vulnerability curve in farming-pastoral ecotone in Northern China, we calculated the exceeded probability of yield loss ratio for each evaluation

unit. With fixed yield loss ratio levels (spring maize yield loss ratio≥0.2, spring maize drought yield loss ratio≥0.3, spring maize yield loss ratio≥0.4 and spring maize yield loss ratio≥0.5), the probability distributions of yield loss ratio are shown in Fig. 12. Seen from the whole study area, under different yield loss ratio levels, the middle part and most region of west part were high risk areas of drought hazard with the upper limit of probability under yield loss ratio levels from 0.2 to 0.5 being 1, 1, 0.8 and 0.5

respectively. This meant the drought hazard in these regions would result in yield loss ratio of 0.2 and 0.3 almost every year. And the drought hazard with yield loss ratio of 0.5 would occur at least every two years. Comparatively, the east part within the rainfall isoline from 500mm to 600mm and the southwest edge of the west part was areas with low probability of drought hazard. The upper limit of probability under 4 hazard levels from 0.2 to 0.5 was 0.8, 0.5, 0.2 and 0.2. So this part was more likely to meet

drought hazard with yield loss ratio smaller than 0.2. And the probability of drought hazard with yield loss ratio of 0.4 was every five years.

In general, the probability distribution of spring maize yield loss ratio was similar with the probability distribution of drought hazard intensity index. The risk of yield loss ratio dropped from arid to humid region. Because physical vulnerability curves of all three parts showed reduction-effect of

drought hazard, the probability of yield loss ratio was slight lower than that of drought hazard intensity index.

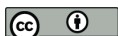



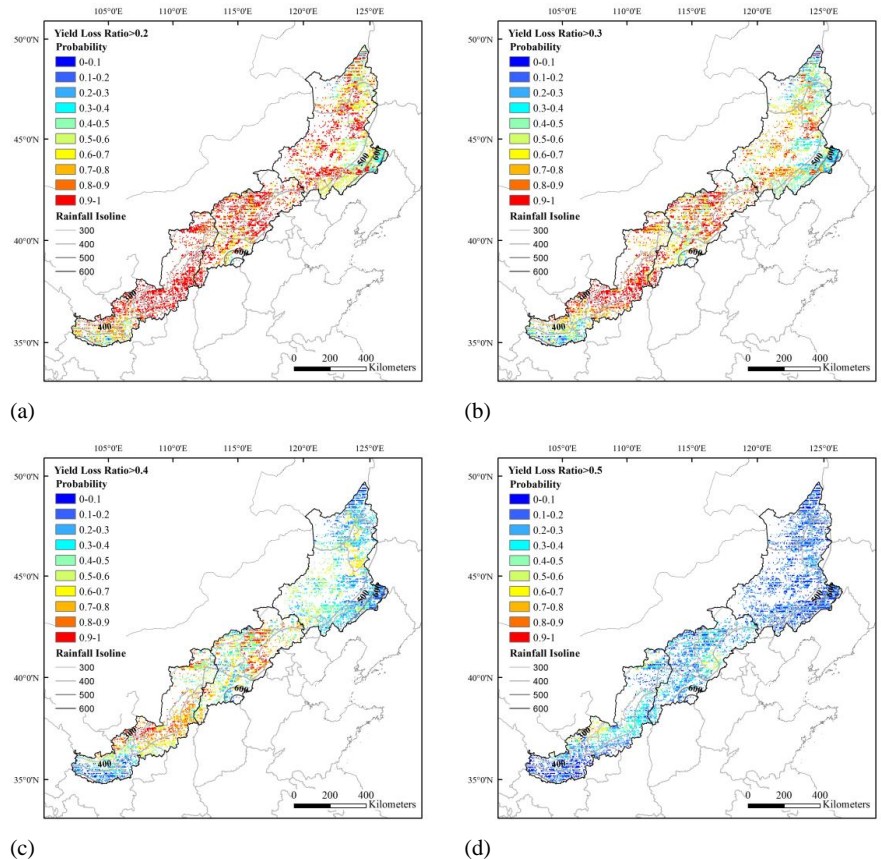

(a)      (b)

(c)      (d)

**Figure 12: Probability distribution of spring maize yield loss ratio for different yield loss levels (a) Yield loss ratio≥0.2; (b) Yield loss ratio≥0.3; (c) Yield loss ratio≥0.4; (d) Yield loss ratio ≥0.5.**

## 4 Discussion

Since the disaster risk are the function of hazard factor, physical vulnerability, exposure and the disaster

reduction capacity of hazard-affected body, so based on the assumption that all maize was exposed to

drought hazard, the drought hazard intensity index was calculated through EPIC model and the physical

vulnerability curve was established for different parts of farming-pastoral ecotone in Northern China.

The risk assessment of drought hazard in study area was discussed from three aspects: The first is the risk

assessment of drought hazard intensity index on spring maize. For the spatial distribution of drought

hazard intensity index, it had the negative correlation with precipitation. For most years, 400mm rainfall

isoline was approximately consistent with drought hazard intensity index of 0.5. For regions with rainfall

isoline less than 400mm, drought hazard intensity index here was usually larger than 0.5, while for



regions with rainfall isoline higher than 400mm, drought hazard intensity index was usually smaller than

0.5. For the time series of drought hazard intensity index, the time variation of drought hazard intensity
index was consistent with the interannual variation of precipitation. For the east part, the interannual
variation of precipitation was larger and presented the tendency of worsening drought, so the standard
deviation was higher. But for most west part, the situation of drought was relatively stable and the
interannual variation of precipitation was smaller, so lower standard deviation of drought hazard

intensity index was showed. Also, for most regions in the study area, the drought hazard intensity index
presented an increasing tendency throughout years. Drought hazard with sever degree was spreading
from west to northeast in recent years. For the probability of drought hazard intensity index, it showed a
tendency of decreasing from southwest to northeast and the tendency of increasing from southeast to
northwest alone the distribution of rainfall isoline. The second is the physical vulnerability assessment

based on physical vulnerability curve. For three parts of farming-pastoral ecotone in Northern China,
similar physical vulnerability curves were got. All of them showed the reduction effect of drought hazard
intensity index. The third is the risk assessment of yield loss ratio calculated from physical vulnerability
curve. Adjusted by the physical vulnerability curve of spring maize, the fluctuation of yield loss ratio was
smaller compared with drought hazard intensity index. Meanwhile, the increasing tendency of yield loss

ratio was slowed down and the probability of yield loss ratio was becoming lower. This meant because of
the physical vulnerability, the capacity of spring maize to resist and adapt to drought was raised.

The risk assessments showed the farming-pastoral ecotone in Northern China is a region with high
risk of agricultural drought and high sensitivity to climate change. Three different parts showed different
spatial and temporal distribution of drought hazard intensity index and yield loss ratio. To better adapt to

drought, measurements can be taken based on the risk assessment in this study: to reduce the drought
hazard intensity, the planting environment can be changed like improving the ability of irrigation or
changing soil property through fertilization and other tillage methods. To reduce physical vulnerability
of crops to agricultural drought, improved varieties of crops can be developed to promote
drought-enduring and drought resisting crops. To reduce crop's exposure to drought, planting structure

can be adjusted during the planting process.

The uncertainty of this study mainly comes from the simulation of EPIC model and the construction
of physical vulnerability curve. For EPIC model, the uncertainties are from the model itself and input
data like meteorological data, soil data and field management data. For the construction of physical



vulnerability curve, the uncertainty is mainly due to the limitation of selected sceneries.

The calculation of physical vulnerability curve to agricultural drought proposed in this method provides a probability to assess drought risk quantitatively. Compared with the previous method, a more accurate drought hazard intensity index was added. According to constructive factors of disaster risk, under the condition of total exposure, the risk assessment was conducted from drought hazard intensity index, physical vulnerability and the yield loss ratio calculated from physical vulnerability curve, which

gave a synthetically assessment on risk of physical vulnerability to agricultural drought of farming-pastoral ecotone in Northern China.

## 5 Conclusion

This study proposed a method to calculate physical vulnerability curve based on the drought hazard intensity index and yield loss ratio from EPIC model. The genetic parameters of spring maize were first

calculated according to the statistical yearbook. Then a water-deficit experiment on different growth stage was conducted to get the yield loss contribution rate. And the drought hazard intensity index was calculated from the daily water stress and yield loss contribution rate for different growth stages. After this, the yield loss ratio was got from the difference of yield with two different simulated scenarios with EPIC model (One was sufficient irrigation and the other one was no irrigation). Then sites under different

drought hazard intensity were selected. A Logistc model was used to build the relationship between hazard and loss and simulate the physical vulnerability curve. Based on the function of disaster risk, under the condition of totally exposure, the risk assessment of agricultural drought of farming-pastoral ecotone in Northern China was conducted from the drought hazard intensity index and physical vulnerability curve. Seen from drought hazard intensity index, the risk of agricultural drought

represented negative correlation with the precipitation. The intensity of drought hazard kept rising for the past 46 years and drought hazard with sever extent was spreading from southwest to northeast gradually. The probability distribution of drought hazard intensity index decreased from southwest to northeast and increased from southeast to northwest alone the rainfall isoline. For the physical vulnerability curve, its reduction effect in three parts of farming-pastoral ecotone in Northern China helped reduce drought

hazard vulnerability on spring maize. For the risk of yield loss ratio based on physical vulnerability curve, the probability was lower compared with the drought hazard intensity index which shows the





capacity of spring maize to resist drought and its adaptation to drought. Overall, the farming-pastoral

ecotone in Northern China is highly sensitive and very fragile to climate change because of its location

with several different transitional zones. Risk assessment of physical vulnerability to agricultural drought

on this region can help people have a better understanding of physical vulnerability to agricultural

drought and can also promote measurements from different fields to adapt to the climate change.

*Acknowledgments.*   This work was supported by a grant entitle ''Study on Agricultural Drought Risk

Formation Mechanism of the Rain-fed Agricultural Typical Area in China'' (41001059) from the

National Science and Technology Foundation. We also thank China Meteorological Administration

(CMA) for data sharing.

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
