# Peer review of "The drought risk of maize in the farming-pastoral ecotone in Northern China based on physical vulnerability assessment"

_Natural Hazards and Earth System Sciences, 2016_

## Short Comment (SC1) · 6 Aug 2016

This paper presents study the drought risk of maize in the farming-pastoral ecotone in Northern China based on physical vulnerability assessment, it was written well, especially the authors did clearly explain the the physical vulnerability curve by building the relationship between drought hazard intensity index and yield loss rate. Importantly, the risk assessment of agricultural drought has been clearly explained. I suggest that it is accepted for publication.

---

## Short Comment (SC2) · 14 Aug 2016

Manuscript Review of nhess-2016-204: Study on the drought risk of maize in the farming-pastoral ecotone in Northern China based on physical vulnerability assessment

I would recommend that this manuscript be published after moderate revision. Please find my comments below.

General comments This paper presents a study on the drought risk of maize in the farming-pastoral ecotone in Northern China. The novelty of the work is to conduct a physical vulnerability curve based on the relationship between drought hazard intensity index and yield loss rate. The study is generally well organized and presented. However, there are several issues which need attention before publication. 1) In the abstract, the authors should offer some quantitative results and conclusions. 2) The description of ecotone in 2.1.1 should be shortened and most of the section should be moved into the section of introduction. 3) China meteorological data sharing service system of China only offers sunshine hours. So, how to transfer the sunshine hours to global radiation? 4) The authors should offer the genetic parameters of maize used in the EPIC model for the three sites in Fig.3. Moreover, please provide the station name of six validation sites. 5) Please provide sowing date of maize, planting density of maize, fertilization amount used in running EPIC model at three representing sites under sufficient and no irrigation conditions. 6) The linear regression curve seemed more appropriate to fit the data than the logistic curve. So, why you select the logistic curve as the physical vulnerability curve? 7) The discussion section should be strengthened by comparison with previous studies, including the impact of drought on spring maize in the farming-pastoral ecotone, the measures used to adapt to climate change, etc. Moreover, please have a native speaker to improve the English of the text. Therefore, I would recommend that this manuscript be published after moderate revision. Other minor comments I suggested that the title should be changed into "The drought risk of maize in the farming-pastoral ecotone in Northern China based on physical vulnerability assessment". P1L10: Make "4" as superscript. P1L21: What does magnify and reduce function mean? P1L22-23: Delete the sentence because it is obvious. P2L31: Change "response to" into "tackling". P2L44-L45: The references should be listed in chronological order. P24L536: Missing the volume and page number of the publication. P3L67: Change "Uzielli et al. (Uzielli et al., 2008)" into "Uzielli et al. (2008)". P3L70: Change "Douglas (Douglas, 2007)" into "Douglas (2007)". P3L73-L74: The references should be listed in chronological order. P3L75: Change "factor" into "factors". P3L77: Change "Wang et al. (Wang et al., 2013)" into "Wang et al. (2013)". P5L158: Change the caption of Table 1 into "Meteorological, soil and relative agricultural data". P7L162: "CH", "CV", "CE" should be consistent with Eq.1. P10L221ïijŽ delete "to" before "the

water stress". P11L48: change "represent" into "representive". Table 2: change "filling" into "grain-filling". P12L270: Change "Wang et al. (Wang et al., 2015)" into "Wang et al. (2015)".

———————————————

---

## Referee Comment (RC1) · Anonymous Referee #1 · 15 Aug 2016

In this manuscript, the risk assessment of agricultural drought was conducted and investigated using the drought hazard intensity index and physical vulnerability curves. The manuscript is overall well-written. However, I believe the manuscript cannot be accepted for publication until the following concerns are properly resolved: 1.In case study, the EPIC model was used to predict daily water stress. However, the EPIC model was not seriously introduced at all. An independent section and the corresponding literature reviews should be added to provide sufficient information of EPIC. 2.Most figures (Figures 1, 5-12) are in poor resolution. Please enhance the quality of figures. 3.I believe most people know the location of China but I suggest adding a map of China along with the studies area in Figure 1. 4.How do you define "Northern China"? As

far as I know, some parts of "Northern China" were not included in this study. Proper justifications should be addressed in the next revision. 5.In conclusion, what will be message to the general public by the proposed work? What could be the limitation(s) of the proposed work? What might be improved in the future based on the proposed work? The conclusion should be substantially revised accordingly.

---

## Referee Comment (RC2) · Anonymous Referee #2 · 18 Oct 2016

The paper presents a case of study of the drought risk of maize in Northern China based on physical vulnerability assessment. The physical vulnerability curve was constructed from the relationship between drought hazard intensity index and yield loss rate. The risk assessment of agricultural drought was conducted from the drought hazard intensity index and physical vulnerability curve. Drought hazard intensity index estimation is based on the daily water stress from EPIC model and yield loss contribution rates for different growth stages. Based on the distribution of drought hazard intensity index, the drought hazard intensity index in different regions was analyzed. Then, the yield loss ratio was obtained from the difference of yield with two different scenarios (sufficient irrigation and no irrigation). A Logistc model was used to simulate
the physical vulnerability curve of crop from the relationship between hazard and loss. According to the physical vulnerability curve, both the physical vulnerability assessment and risk assessment of yield loss ratio were analyzed. The topic of the paper is interesting and the manuscript is well written. I proposed the publication after some minor revisions.

General Comments: 1)It is important to include in some part of the introduction the differences about hazard, vulnerability and risk, because sometimes are used indistinctly. For example, the author can use as basis the terminology used by UNISDR (https://www.unisdr.org/we/inform/terminology). 2)It can be seen in figure 5 that the drought hazard intensity index has a cyclic behavior with a return period of 20 years aprox. How is this considered in the risk assessment? 3)It could be illustrative to include in the conclusions the weaknesses and limitations of the approach. 4)It is difficult to read some figures because the size of labels is too small.

---

## Author Comment (AC1) · 6 Nov 2016

Response to RC1:

In this manuscript, the risk assessment of agricultural drought was conducted and investigated using the drought hazard intensity index and physical vulnerability curves.

The manuscript is overall well-written. However, I believe the manuscript cannot be accepted for publication until the following concerns are properly resolved:

1. In case study, the EPIC model was used to predict daily water stress. However, the EPIC model was not seriously introduced at all. An independent section and the corresponding literature reviews should be added to provide sufficient information of EPIC.

**Comments of reviewer are very valuable. A short description about EPIC and the corresponding literature reviews has been added in Section 2.2.1.**

**EPIC model is a field-scale crop model, which is capable of simulating daily crop growth, calculating crop yield under various climate and environment conditions and performing long-term simulations for hundreds of years (Gassman et al., 2005). In recent years, EPIC model has been applied in different fields, including climate change (Izaurralde et al., 2012; Rinaldi and De Luca, 2012), simulation of crop yields (Pumijumnong and Arunrat, 2013; Xiong et al., 2014) and drought disaster risk assessment (Jia et al., 2012; Wang et al., 2013b). In previous study (Wang et al., 2015),a new method was proposed to determine drought hazard intensity index based on the daily water stress from EPIC model and yield loss contribution rates for different growth stages. In this study, the risk assessment of agricultural drought was conducted from the physical vulnerability curve.**

2. Most figures (Figures 1, 5-12) are in poor resolution. Please enhance the quality of figures.

**I am sorry for the poor resolution of figures. Because of the limitation of WORD, figures with high quality will cause the file become too large. All the figures with high resolution will be uploaded as the attachments for the final version.**

3. I believe most people know the location of China but I suggest adding a map of China along with the studies area in Figure 1.

**A map of China along with the studies area has been added in Figure 1.**

4. How do you define "Northern China"? As far as I know, some parts of "Northern China" were not included in this study. Proper justifications should be addressed in the next revision.

**Generally speaking, there is no precisely definition of Northern China. But a geographical dividing line between northern and southern China which is named the Huai River–Qin Mountains Line is often used to define northern and southern China. However, in this study, the study area is the farming-pastoral ecotone in Northern China. There exist many different definitions about the farming-pastoral ecotone in northern China. In general, it is located at the north part of China with the rainfall isoline changing from 300mm to 400mm, annual precipitation change ranging from 15% to 30% and dryness changing from 1.0 to 2.0 (Zhao et al., 2002).**

5. In conclusion, what will be message to the general public by the proposed work? What could be the limitation(s) of the proposed work? What might be improved in the future based

on the proposed work? The conclusion should be substantially revised accordingly.

**We thank the reviewers for the suggestions. Some revision has been made in discussion and conclusion part.**

**The risk assessments showed the farming-pastoral ecotone in Northern China is a region with high risk of agricultural drought and high sensitivity to climate change. Three different parts showed different spatial and temporal distribution of drought hazard intensity index and yield loss ratio. Drought is one of the most manifestations of climate variability in this region and severe droughts are becoming more frequently in recent years. To better adapt to drought, measurements can be taken based on the risk assessment in this study: to reduce the drought hazard intensity, the planting environment can be changed like improving the ability of irrigation or changing soil property through fertilization and other tillage methods. To reduce physical vulnerability of crops to agricultural drought, improved varieties of crops can be developed to promote drought-enduring and drought resisting crops. To reduce crop's exposure to drought, planting structure can be adjusted during the planting process.**

**The uncertainty of this study mainly comes from the simulation of EPIC model and the construction of physical vulnerability curve. For EPIC model, the uncertainties are from the model itself and input data like meteorological data, soil data and field management data. For the construction of physical vulnerability curve, the uncertainty is mainly due to the limitation of selected sceneries.**

**For the further study, a larger study area including south and north part of China will be selected to better assess drought risk and describe the impact of climate change to agriculture along different latitudes.**

---

## Author Comment (AC2) · 6 Nov 2016

Response to RC2:

The paper presents a case of study of the drought risk of maize in Northern China based on physical vulnerability assessment. The physical vulnerability curve was constructed from the relationship between drought hazard intensity index and yield loss rate. The risk assessment of agricultural drought was conducted from the drought hazard intensity index and physical vulnerability curve. Drought hazard intensity index estimation is based on the daily water stress from EPIC model and yield loss contribution rates for different growth stages. Based on the distribution of drought hazard intensity index, the drought hazard intensity index in different regions was analyzed. Then, the yield loss ratio was obtained from the difference of yield with two different scenarios (sufficient irrigation and no irrigation). A Logistc model was used to simulate the physical vulnerability curve of crop from the relationship between hazard and loss.

According to the physical vulnerability curve, both the physical vulnerability assessment and risk assessment of yield loss ratio were analyzed. The topic of the paper is interesting and the manuscript is well written. I proposed the publication after some minor revisions.

General Comments:

1) It is important to include in some part of the introduction the differences about hazard, vulnerability and risk, because sometimes are used indistinctly. For example, the author can use as basis the terminology used by UNISDR (https://www.unisdr.org/we/inform/terminology).

**We thank the reviewers for the suggestions. In the revision, the differences about hazard, vulnerability and risk have been added in the introduction part. The hazard is called a dangerous phenomenon, substance, human activity or condition that may cause loss of life, injury or other health impacts, property damage, loss of livelihoods and services, social and economic disruption, or environmental damage (UNISDR, 2009). The risk is defined as the combination of the probability of an event and its negative consequences (UNISDR, 2009). Initially, vulnerability was defined as the human response to hazard events (Blaikie and Cannon, 1994;FAO, 2001). Gradually, vulnerability is added with some new meanings including the different systems of human society responding to hazard, the interaction process of multi-factors like nature, society, economy and environment (UNDP, 2004) , the sensitivity or susceptibility to hazards and the capacity to cope and adapt to hazards (IPCC, 2014).**

2) It can be seen in figure 5 that the drought hazard intensity index has a cyclic behavior with a return period of 20 years aprox. How is this considered in the risk assessment?

**In Figure 5, there exist two extreme drought hazards in 1980 and 2000 with the average of drought hazard intensity index larger than 0.8. For other years, areas with the drought hazard intensity index larger than 0.5 mainly centered in the west part. A cyclic behavior with a return period of 20 years aprox appears from 1965 to 2000. It show the periodicity of the extreme drought event and can be used as the reference for the prediction of drought hazard.**

3) It could be illustrative to include in the conclusions the weaknesses and limitations of the approach.

**We thank the reviewers for the suggestions. Some revision has been made in discussion and conclusion part.**

**The uncertainty of this study mainly comes from the simulation of EPIC model and the construction of physical vulnerability curve. For EPIC model, the uncertainties are from the**

**model itself and input data like meteorological data, soil data and field management data. For the construction of physical vulnerability curve, the uncertainty is mainly due to the limitation of selected sceneries.**

**For the further study, a larger study area including south and north part of China will be selected to better assess drought risk and describe the impact of climate change to agriculture along different latitudes.**

4) It is difficult to read some figures because the size of labels is too small.

**In the revision, the size of labels have been adjusted to be larger.**

---

## Author Comment (AC4) · 6 Nov 2016

Response to SC2:

Manuscript Review of nhess-2016-204: Study on the drought risk of maize in the farming-pastoral ecotone in Northern China based on physical vulnerability assessment I would recommend that this manuscript be published after moderate revision. Please find my comments below.

General comments

This paper presents a study on the drought risk of maize in the farming-pastoral ecotone in Northern China. The novelty of the work is to conduct a physical vulnerability curve based on the relationship between drought hazard intensity index and yield loss rate. The study is generally well organized and presented.

However, there are several issues which need attention before publication.

1) In the abstract, the authors should offer some quantitative results and conclusions.

**Comments of reviewer are very valuable. Some revisions have been made in abstract.**

2) The description of ecotone in 2.1.1 should be shortened and most of the section should be moved into the section of introduction.

**The description of ecotone in 2.1.1 has been modified and moved to introduction part.**

3) China meteorological data sharing service system of China only offers sunshine hours. So, how to transfer the sunshine hours to global radiation?

**Daily solar radiation information was recorded in 27 stations. The daily solar radiation data for the remaining stations were estimated based on the sunshine duration data using the Angstrom-Prescott model (Angstrom, 1924;Prescott, 1940)**

4) The authors should offer the genetic parameters of maize used in the EPIC model for the three sites in Fig.3. Moreover, please provide the station name of six validation sites.

**The genetic parameters of maize for three sites have been listed below. The six validation sites are Chifeng, Tongliao, Zhangjiakou, Jining, Guyuan and Dingxi.**

| Parameter name | Meaning of parameter | Baicheng | Datong | Yulin |
|---|---|---|---|---|
| WA | Energy- biomass conversion factor | 37 | 39 | 44 |
| HI | Harvest index | 0,6 | 0,5 | 0,65 |
| TB | The most suitable temperature for crop growth (℃) | 25 | 25 | 25 |
| TG | The lowest temperature for crop growth (℃) | 5 | 5 | 5 |
| DMLA | The maximum potential leaf area index | 7 | 7 | 7 |
| DLAI | The ratio of LAI downward stage accounted for the growing season | 0,18 | 0,2 | 0,15 |
| DLP1 | Crop area growth curve parameter 1 | 15,05 | 15,05 | 15,05 |
| DLP2 | Crop area growth curve parameter 2 | 50,95 | 50,95 | 50,95 |
| RLAD | Leaf area index decreasing parameter | 0,1 | 0,1 | 0,1 |

5) Please provide sowing date of maize, planting density of maize, fertilization amount used in running EPIC model at three representing sites under sufficient and no irrigation conditions.

**The sowing date of maize is set to be April 25th (Baicheng), April 15th (Datong) and April 10th (Yulin) separately based on the Chinese Planting Information Network (http://www.seedchina.com.cn/). Planting density and fertilization amount are set to be automatic mode in EPIC model.**

6) The linear regression curve seemed more appropriate to fit the data than the logistic curve. So, why you select the logistic curve as the physical vulnerability curve?

**Here we chose logistic curve instead of a straight line to simulate physical vulnerability curve because logistic curve can be used to describe the drought hazard intensity dependent biphasic effect of maize physical vulnerability to drought disaster. At the beginning and the end of the curve, the slope is small. This means for both low hazard intensity and high hazard intensity, the increasing of drought hazard intensity has relatively small impact on the yield loss ratio. However, for the middle part of the curve, the slope is large. This means for middle hazard intensity, the increasing of drought hazard intensity will have larger impact on the yield loss ratio. In this study, restricted by the meteorological data, it was hard to include every different meteorological scenery in theoretical like extreme drought (H = 1) or no drought (H = 0) to simulate a real physical vulnerability curve to drought hazard of spring maize. In addition, errors from meteorological data and the model itself might also have impacts on simulation results. So considering the accuracy of the input data and some uncertainties during the calculation process, the simulated drought physical vulnerability curve of spring maize for each part was satisfied.**

7) The discussion section should be strengthened by comparison with previous studies, including the impact of drought on spring maize in the farming-pastoral ecotone, the measures used to adapt to climate change, etc. Moreover, please have a native speaker to improve the English of the text. Therefore, I would recommend that this manuscript be published after moderate revision.

**We thank the reviewers for the suggestions. Some revisions have been made in disscussion part. The risk assessments showed the farming-pastoral ecotone in Northern China is a region with high risk of agricultural drought and high sensitivity to climate change. Three different parts showed different spatial and temporal distribution of drought hazard intensity index and yield loss ratio. Drought is one of the most manifestations of climate variability in this region and severe droughts are becoming more frequently in recent years. To better adapt to drought, measurements can be taken based on the risk assessment in this study: to reduce the drought hazard intensity, the planting environment can be changed like improving the ability of irrigation or changing soil property through fertilization and other tillage methods. To reduce physical vulnerability of crops to agricultural drought, improved varieties of crops can be developed to promote drought-enduring and drought resisting crops. To reduce crop's exposure to drought, planting structure can be adjusted during the planting process.**

Other minor comments I suggested that the title should be changed into "The drought risk of maize in the farming-pastoral ecotone in Northern China based on physical vulnerability assessment".

P1L10: Make "4" as superscript.

P1L21: What does magnify and reduce function mean?

P1L22-23: Delete the sentence because it is obvious. P2L31: Change "response to" into "tackling".

P2L44-L45: The references should be listed in chronological order.

P24L536: Missing the volume and page number of the publication.

P3L67: Change "Uzielli et al. (Uzielli et al., 2008)" into "Uzielli et al. (2008)".

P3L70: Change "Douglas (Douglas, 2007)" into "Douglas (2007)".

P3L73-L74: The references should be listed in chronological order.

P3L75: Change "factor" into "factors".

P3L77: Change "Wang et al. (Wang et al., 2013)" into "Wang et al. (2013)".

P5L158: Change the caption of Table 1 into "Meteorological, soil and relative agricultural data".

P7L162: "CH", "CV", "CE" should be consistent with Eq.1.

P10L221ïïjŽ delete "to" before "the water stress".

P11L48: change "represent" into "representive".

Table 2: change "filling" into "grain-filling".

P12L270: Change "Wang et al. (Wang et al., 2015)" into "Wang et al. (2015)".

**The reviewers are correct about some minor issues. We have accepted and revised all minor issues (include the words and figures) in the manuscript. And some repetitious part of the manuscript has been cut and refined.**